

# Spatial and temporal population genetic analysis of *Semaprochilodus insignis* (Prochilodontidae), an overexploited fish from the Amazon basin

Ingrid Nunes[1], Kelmer Passos[1], Aline Mourão Ximenes[1,2], Tomas Hrbek[1,3] and Izeni Pires Farias[1]

[1] Departamento de Genética, Laboratório de Evolução e Genética Animal (LEGAL), Universidade Federal do Amazonas, Manaus, Amazonas, Brasil

[2] Programa de Pós-Graduação em Genética, Conservação e Biologia Evolutiva, Instituto Nacional de Pesquisas da Amazônia, Manaus, Amazonas, Brasil

[3] Department of Biology, Trinity University, San Antonio, TX, United States of America

## ABSTRACT

**Background**. *Semaprochilodus insignis* is a migratory fish of commercial and subsistence importance to communities in the Amazon. Despite the high intensity of exploitation, recent studies have not been carried out to assess the genetic status of its stocks.

**Methods**. This study is the first to estimate genetic diversity and to test the existence of spatial and temporal structuring of *S. insignis* through sequencing of the mtDNA control region ($n = 241$) and eight microsatellite loci ($n = 180$) of individuals sampled at 11 sites distributed in the Brazilian Amazon basin.

**Results**. Results for both markers were congruent, revealing a homogeneous genetic diversity in all the sampled locations, in addition to the absence of spatial and temporal genetic structure, indicating that the species forms a large panmictic population in the Brazilian Amazon.

**Discussion**. Although overfishing does not yet appear to have affected the levels of genetic variability of *S. insignis*, signals of reduction of the effective population size and a bottleneck provide an early alert to the effects of overfishing. Thus, the ever-decreasing populations may threaten *S. insignis* in the future. Therefore, it is hoped that the results of this study may contribute to the elaboration of management plans or any other measures that aim at the management and conservation of this species of great importance for the Amazon basin.

## INTRODUCTION

In the Amazon, fishing is one of the most important activities for a large portion of the population in terms of food, commerce, income and leisure, especially for populations that reside on the margins of large and medium-size rivers (*Santos & Santos, 2005*).

Corresponding author
Izeni Pires Farias,
izeni@evoamazon.net

Between six and 12 species of fish make up over 80% of landings at the main ports in the region (*Ruffino et al., 2006*), and among them one of the most fished species is the jaraqui *Semaprochilodus insignis* (Jardine, 1841). Due to its comparatively low commercial value, *S. insignis* is the main fish consumed by low-income urban population (*Araújo-Lima & Hardy, 1987*; *Faria Jr & Batista, 2019*). Its capture is concentrated in the central Amazon, where it accounts for 93% of the entire fish landing (*Barthem & Goulding, 2007*). Studies carried out with the species in the 1990s (*Ribeiro, De & Petrere-Jr, 1990*) detected a signal of possible overfishing. This situation was also reported by *Batista (1998)* and *Batista et al. (2012)*, who suggested that the greatest exploitation occurs with smaller individuals, which may be an indication that the stock has been strongly affected by fishery exploitation. The smaller size of individuals has also been noticed by fishermen who, according to *Barthem & Goulding (2007)*, have begun to limit their catch to adult fish in hopes that stocks will recover. However, actual harvest (as tons of fish landed) only increased (*Neto & Dias, 2015*), while at the same time the size of individual fish decreased (*Batista et al., 2012*). *Semaprochilodus insignis* also has a strong cultural significance in the central Amazon where a popular limerick says "Quem come jaraqui não sai mais daqui"—"Whoever eats jaraqui no longer will leave from here". This reflects how appreciated this species is in the cuisine of the low income classes of the Amazon region for whom *S. insignis* is often the only or major source of animal protein. *Semaprochilodus insignis* also occupies the third position among the top ten species or group of species with the highest average production by the national continental fisheries during the period from 1995 to 2010 (*Neto & Dias, 2015*).

*Semaprochilodus insignis* is a medium-sized fish, reaching up to 35 cm in length, with caudal and anal fins adorned with dark diagonal bands interspersed with orange yellow bands. The species occurs in the central and western portions of the Amazon basin and its main tributaries. Its distribution includes Brazil, Colombia, Ecuador, Guyana and Peru. The majority of jaraqui spawn for the first time during the first years of life, and are r-strategists, producing thousands of offspring (*Ribeiro, De & Petrere-Jr, 1990*; *Vazzoler, Amadio & Caraciolo-Malta, 1989*). *Semaprochilodus insignis* migrates twice a year (*Araújo-Lima & Hardy, 1987*; *Goulding, 1980*), carrying out seasonal migrations for feeding and reproduction. The species migrates hundreds of kilometers, and the annual migrations in central Amazon may extend for 1000–1300 km with annual maximum upstream-displacements of 300 km in the white-water rivers (*Araújo-Lima, Higuchi & Barrella, 2004*; *Granado-Lorencio et al., 2005*; *Ribeiro, De & Petrere-Jr, 1990*; *Vazzoler, Amadio & Caraciolo-Malta, 1989*). The first migration is for reproduction at the beginning of the flood. The species leaves nutrient-poor tributaries (black and clear waters) for downstream turbid-water rivers (white water) in compact and large schools of adults and subadults to spawn at the encounter with the main river, where the waters are turbid and rich in nutrients. Eggs and early larvae are carried passively by currents to nursery grounds which are rich in food and provide refuge from predators (*Lowe-McConnell, 1987*). The second migration is the dispersion, which occurs in the middle of the flood season, when the fish, once again, leave rivers that are poor in nutrients, they migrate to rivers rich in food, successively entering and leaving other tributaries during the low water season. At this dispersal migration the fishes ascend the Amazon and the tributaries at 11.5 km day,

successively entering and leaving tributaries until the onset of the low water season (*Ribeiro, De & Petrere-Jr, 1990*). It is during these migrations that the species is intensively fished.

In the year 2000, the Ministry of the Environment (MMA) classified the *S. insignis* as overexploited or threatened with overexploitation and encouraged the drafting of a management plan with the aim of recovering its stocks. Therefore, the species is included in Annex M of Normative Instruction MMA No. 05/2004, which lists overexploited Brazilian species or those threatened with overexploitation. *Semaprochilodus insignis* is also listed in IBAMA's (Brazil's Environmental Protection Agency) list of threatened species as overexploited or threatened with overexploitation. The species is a flagship species that plays an important role as an ecosystem engineer (*Goulding et al., 2018*), and its current overexploitation and commercial importance (*Batista, 1998*; *Vieira, 2003*) makes it imperative that detailed biological and genetic studies are carried out, so that effective conservation and management programs can be implemented. Given the above, the characteristics of the species, the great demand and the increasing level of fishing effort used to capture it, *Neto & Dias (2015)* reinforces the situation that *S. insignis* are overfished.

The present study examines the population genetics of *S. insignis* by sequencing the control region of the mitochondrial DNA and genotyping eight microsatellite loci from individuals collected from 11 localities along the main channel of the Amazon River and its tributaries, which include rivers with different water types, such as black water (Negro River), clear water (Tapajós River) and white water (Solimões-Amazon Rivers). Considering the migratory patterns of this r-strategist species, we expect it to form a single large panmictic population despite being found in a diversity of water types and forming different migratory groups throughout the Amazon basin. In addition, considering the historical situation of overfishing of the species, we test whether populations already show signs of population size reduction in their genetic stocks.

## MATERIALS & METHODS

### Field sampling and laboratory protocols

In total, 241 specimens of the *Semaprochilodus insignis* were sampled from small artisanal fishing boats at 11 localities in the Amazonian basin (Fig. 1, Table 1), from Tabatinga to Santarém (west–east) along the main channel of the Amazon and in its northern and southern tributaries. These localities include black water rivers (Negro River), clear water rivers (Tapajos River) and white water rivers (localities in the mainstream of Solimões-Amazon Rivers and the tributaries Juruá/Purus). Samples of pectoral fins were collected and stored in 95% ethanol. Collections from different years were possible for four sites, Tabatinga (2007–2008), Purus (2006–2007), Santarém and Itaituba (2006–2008), enabling to test possible temporal variations in the genetic structure of *S. insignis*. We were also able to collect samples from geographically distant migrating aggregations, enabling us to test for divergence of migratory aggregations. All individuals were captured and sampled under license granted by the Instituto Brasileiro do Meio Ambiente e dos Recursos Naturais Renováveis (IBAMA/SISBIO permit #11325-1). Sampling collection was undertaken in accordance with the ethical recommendations of the Conselho Federal de Biologia (CFBio; Federal Council of Biologists), Resolution 301 (December 8, 2012).
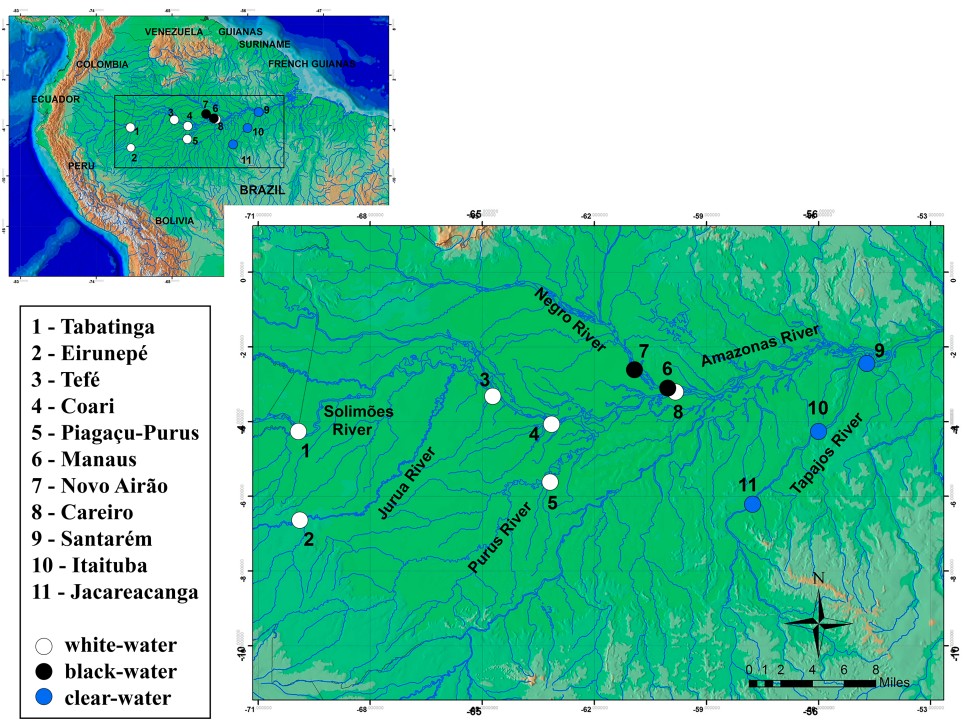

**Figure 1 Sampling locations for *Semaprochilodus insignis*.** Types of water in river drainages are represented by white (white water), black (black water) and blue (clear water) circles.

**Table 1 Sample size per locality and year of sampling.**

| | | Markers | |
|---|---|---|---|
| **Localities (year)** | **Localities Code** | **Control Region (per year)** | **Microsatellite (per year)** |
| 1. Tabatinga (2007 & 2008) | Tab | 18 (12 + 6) | 17 (11 + 6) |
| 2. Eirunepé (2008) | Eir | 32 | 20 |
| 3. Tefé (2007) | Tef | 12 | 12 |
| 4. Coari (2007) | Coa | 10 | 7 |
| 5. Piagaçu-Purus (2006 & 2007) | PPu | 36 (30 + 6 ) | 19 (9 + 10) |
| 6. Manaus (2008) | Mao | 21 | 20 |
| 7. Novo Airão (2008) | NAi | 23 | 21 |
| 8. Careiro (2008) | Car | 12 | 9 |
| 9. Santarém (2006 & 2008) | San | 32 (14 + 18) | 20 (12 + 8) |
| 10. Itaituba (2006 & 2008) | Itai | 28 (11 + 17) | 18 (8 + 10) |
| 11. Jacareacanga (2008) | Jac | 17 | 17 |
| **Total** | | **241** | **180** |

For the spatial analysis, 241 individuals were sequenced for mitochondrial DNA control region and 180 individuals were genotyped for eight microsatellite loci (Table 1). The same individuals that were genotyped were also sequenced for the control region. However, a

few sequenced individuals could not be genotyped. A subset of this database, composed of samples collected in the same location in different years, was used to perform the temporal analyses, being composed of 114 individuals for the control region of mitochondrial DNA and 74 individuals for eight microsatellite loci.

From each sample, the total DNA was extracted following a standard proteinase K, phenol–chloroform extraction protocol (*Sambrook, Fritsch & Maniatis, 1989*). The mtDNA control region was amplified *via* the polymerase chain reaction (PCR) using the primers F-TTF (5′-GCCTAAGAGCATCGGTCTTGTAA-3′) and 12SR5 (5′-GGCGGATACTTGCATGT-3′) obtained from *Sivasundar, Bermingham & Orti (2001)* and *Hrbek & Farias (2008)*, respectively. The PCR reactions were carried out in 15 µL total volume and consisted of 1 µL of total genomic DNA (50–100 ng/µL), 2 mM each primer, 1X PCR Buffer (2 mM Tris-KCL, pH 8.5), 1.5 mM MgCl2 (25 mM), 1.5 mM dNTPs, 1U/µL Taq DNA polymerase. The PCR condition consisted of 35 cycles of denaturation at 93 °C for 60 s, annealing at 50 °C for 40 s, extension at 72 °C for 90 s and final extension was performed at 72 °C for 5 min. Subsequent to PCR amplification, the PCR products were purified using exo-sap (Fermentas), and sequence reactions were carried out using the BigDye TM Terminator V3.1 Cycle Sequencing Kit (Life Technologies) following the manufacturer's instructions. Sequences were resolved using the ABI 3130XL sequencer (Life Technologies) (*Hall, 1999*) and manually verified in the software BioEdit (*Hall, 1999*). Alignment was carried out in BioEdit using the software ClustalW (*Thompson, Higgins & Gibson, 1996*) under default conditions.

Eight microsatellite loci characterized by *Passos et al. (2010)* were used in this study. Seven of them were amplified in three multiplex PCRs, and the selection of multiplex systems was performed by combining allele sizes so that there was no overlap in the microsatellite ranges (Table S1). The total volume of PCR reaction was of 15 µL for a multiplex system consisting of three microsatellite loci, 14 µL for a multiplex system consisting of two microsatellite loci and 8.5 µL for an individually amplified microsatellite marker. The PCR reaction consisted of 1 µL of genomic DNA (50–100 ng/ µL), 1.5 mM MgCl2 (25 mM), 1.5 mM dNTPs, 1X PCR Buffer (2mM Tris-KCL, pH 8.5), 2 mM for tailed forward primer, 2 mM for reverse primer, 2 mM for primer M13 labeled primer(with FAM fluorescence) and 2.5 U Taq DNA polymerase. Reactions were submitted to the following cycling: hot start at 94 °C for 60 s, followed by 25 cycles of denaturing at 94 °C for 30 s, annealing at 55 °C for 30 s, and extension at 68 °C for 40 s; labeling step consisted of 20 cycles of denaturing at 94 °C for 20 s, annealing at 53 °C for 30 s, and extension at 72 °C for 60 s; final extension at 72 °C for 30 min. The PCR products were visualized in a 2% agarose gel, dyed with GelRed and photographed using a UV 302 nm transilluminator T26M (BioAgency).

PCR products generated with labeled primers were visualized on an ABI-3500 automatic sequencer (Life Technologies). Allele sizes were scored against a ROX size standard (*DeWoody et al., 2004*) and the results were analyzed in software GeneMapper® version (Life Technologies).

## Data analysis
### Genetic diversity

For the mtDNA control region we estimated genetic diversity as gene diversity ($\hat{H}$), nucleotide diversity ($\Pi$), number of haplotypes and number of segregating sites (S). For the reconstruction of genealogical relationships among individuals of *S. insignis* a haplotype network based on a maximum likelihood phylogenetic tree topology was used in the Haploviewer software (*Salzburger, Ewing & Von Haeseler, 2011*). For microsatellite data, genetic diversity was estimated from averages number of alleles, gene diversity and observed and expected heterozygosities and inbreeding coefficient ($F_{IS}$) for each locality. Furthermore the number of alleles observed and expected heterozygosities, and linkage disequilibrium (LD) between pairs of loci, and Hardy-Weinberg equilibrium (HWE) were calculated for each locus within the localities. All these measures of genetic variability were estimated in the software ARLEQUIN version 3.5 (*Excoffier & Lischer, 2010*). Statistical significance for LD was tested using 95% confidence intervals and 10.000 permutations and for HWE 1000.000 permutations were used, probability values were adjusted with the Bonferroni correction, when necessary (*Rice, 1989*). Considering that number of alleles suffer influence of sample size (*Leberg, 2002*), a rarefaction analysis was implemented in the software HP-Rare (*Kalinowski, 2005*) that solves this problem, allowing to estimates allelic richness, and private alleles. Estimates of heterozygosity are less influenced by the sample size (*Nei & Roychoudhury, 1974*). The software Micro-Checker (*Van Oosterhout et al., 2004*) was used to detect the existence of null alleles and potential genotyping errors.

### Population structure

Using the Molecular Analysis of Variance (AMOVA) (*Excoffier, Smouse & Quattro, 1992*) and pairwise $\Phi_{ST}$ (control region) and $F_{ST}$ (microsatellite) metrics, we tested the hypothesis of overall population structure (all 11 samples), temporal structure (the four localities sampled in two different years each), and structure associated with differences in water chemistry. The statistical significance for all analyzes was assessed using 20.000 permutations after adjusting for multiple comparisons (*Rice, 1989*).

Additional analysis for the delimitation of biological populations was obtained from the microsatellite data using Bayesian analysis implemented in the STRUCTURE *v.* 2.3.4 software (*Pritchard, Stephens & Donnelly, 2000*). We implemented the admixture model which allows individuals to have ancestors from more than one population and a model of correlated allele frequencies, but included no *a priori* information on the existence of population for each groups. We performed 10 independent runs for each K value which ranged from one to 12; we ran 1,000,000 Monte Carlo chains (MCMC) and discarded the first 100,000 as burn in. The most likely K value was estimated using Bayes Factors (*Pritchard, Stephens & Donnelly, 2000*)

The Discriminant Analysis of Principal Components (DAPC) (*Jombart, Devillard & Balloux, 2010*), using adegenet v2.1.1 (*Jombart, 2008*) in R v3.5.2 (*R Development Core Team, 2011*), was also used to investigate the most likely number of genetic clusters present in the dataset. The analysis tests different clusters (k) using the Bayesian Information Criterion (BIC), where the best cluster corresponds to the lowest BIC value and optimal

number of principal component (PCs) to be retained are determined *via* a-score. The method maximizes variation between cluster, minimizing variation within clusters (*Jombart & Collins, 2015*).

Hypothesis of isolation by distance was tested by the Mantel test (*Mantel, 1967*) using $\Phi_{ST}/(1 - \Phi_{ST})$ for the control region and $F_{ST}/(1 - F_{ST})$ for the microsatellites, and the distances in kilometers representing the shortest river path between each pair of populations (*Fetzner & Crandall, 2003*). The Spearman correlation (*Spearman, 1904*) with 20,000 permutations was used to evaluate the statistical significance for both the control region and the microsatellites in ARLEQUIN v3.5 (*Excoffier & Lischer, 2010*).

### Population sizes and bottleneck tests

Considering the situation of overfishing of the species, we estimated the effective population size (Ne) for each population in the software NeEstimator v 2.0 (*Do et al., 2014*), using the LDNe method (*Waples & Do, 2008*) that assumes a random mating model and allele frequencies cut off of 0.02. This analysis estimates the number of individuals who contribute to the sample based on linkage disequilibrium. We also tested whether populations have recently experienced a reduction in effective population size through the following methodologies: (1) we used the software BOTTLENECK (*Piry, Luikart & Cornuet, 1999*), which identifies populations that have experienced a reduction in effective population size through presence of heterozygosity excess, implemented using two different mutation models: the Stepwise Mutation model (SMM) (*Ohta & Kimura, 1973*) and the Infinite Alleles model (IAM) (*Estoup et al., 1995*), and for each one of the models we applied the standardized differences test; and (2) we used the Garza-Williamson index (M-ratio; *Garza & Williamson, 2001*) estimated in ARLEQUIN v3.5 (*Excoffier & Lischer, 2010*) to test if the reduction in the number of alleles ($k$) compared to allelic spread ($r$) is below the critical value 0.68, which indicate that the population has experienced a recent reduction in size, featuring a bottleneck event.

## RESULTS

### Genetic diversity

The control region data were composed of 241 nucleotide sequences with a total of 1,145 base pairs (bp) of which 1,031 sites were monomorphic and 114 variables. The sequences are available at GenBank: ON972834–ON973074. In the 241 individuals collected along the main channel and tributaries, 116 haplotypes were found. The lowest number of haplotypes was observed in the locality of Careiro (seven) and the highest in Purus and Santarém (27) (Table S2). The H4 haplotype had the highest frequency (45 individuals) and geographical distribution, being present in all sampled locations in the Amazon basin. Of the 116 haplotypes, 89 haplotypes were singletons.

The results of the genetic diversity parameters show that gene diversity was 1 in all locations (Table 2). The lowest haplotype diversity was observed in Tabatinga and Novo Airão (0.009) and the highest values in Coari, Piagaçu-Purus, Manaus and Careiro (0.012) (Table 2). The results of the haplotype network show extensive haplotype sharing and
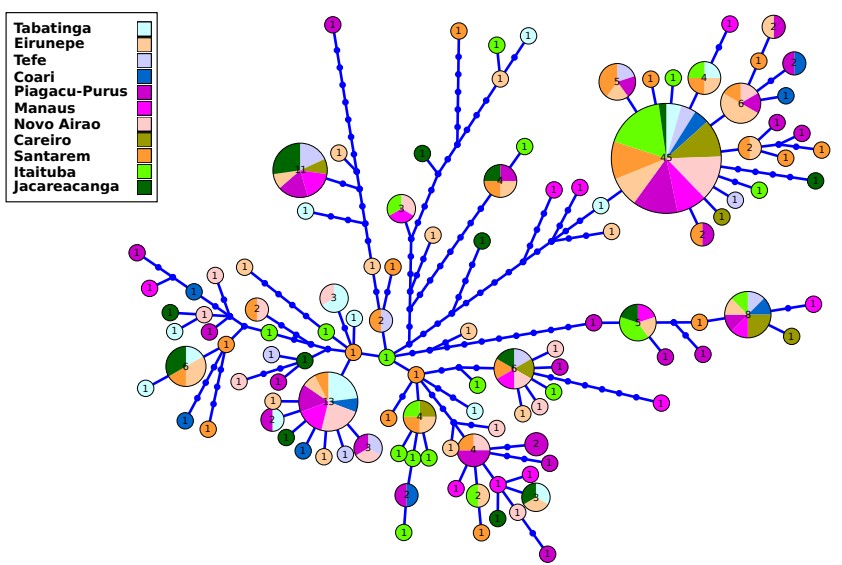

**Figure 2** **Haplotype network of control region from *Semaprochilodus insignis* sampled in the Brazilian Amazon.** Haplotype network showing intense sharing of haplotypes/absence of genetic structure in all sampled locations in the Brazilian Amazon. The size of each circle corresponds to the relative frequency of the haplotype in the entire data set.

several unique haplotypes indicating an absence of genetic structure for the sampled locations in the Brazilian Amazon (Fig. 2).

The analysis based on microsatellite loci of the 180 individuals of *S. insignis* showed 191 alleles in the eight analyzed loci. Matrix of genotypes is available at https://github.com/legalLab/datasets and https://doi.org/10.5281/zenodo.7641680. The number of alleles per locus ranged from 12 (SinC10 locus) to 31 (SinF05 and SinE08 loci) (mean = 23.88 alleles per locus). The average expected heterozygosity(HE) ranged from 0.79 (locus SinB04) to 0.94 (locus SinE08), with an average of 0.88 per locus (Table S3). After Bonferroni correction, no loci deviated significantly from HWE and there was no evidence of linkage disequilibrium. Within locations the average number of alleles ranged from 7.4 in Coari to 15.0 in Jacareacanga. The average values of gene diversity over loci were very similar in all locations with the minimum value in Coari (0.85 ± 0.48) and the maximum value in Eirunepé and Jacareacanga (0.91 ± 0.49). The average allelic richness was also quite homogeneous among locations, ranging from 5.42 in Tefé to 6.09 in Jacareacanga. Observed heterozygosity ranged from 0.74 in Itaituba to 0.85 in Tabatinga, Eirunepé and Tefé, while the expected heterozygosity ranged from 0.87 in Tefé, Coari, Piaguaçu-Purus, Santarém and Itaituba to 0.91 in Jacareacanga (Table 2). The inbreeding coefficient ($F_{IS}$) ranged from 0.021 in Tefé to 0.234 in Manaus, showed statistical significance for eight out of 11 sampled locations (Eirunepé, Coari, Manaus, Novo Airão, Careiro, Santarém, Itaituba and Jacareacanga); it also was significant when the analysis was conducted considering a single and large population. Proportion of private alleles ranged from 0.40 in Tefé and Coari to 0.85 in Jacareacanga. Private alleles values

**Table 2   Main genetic parameters for *Semaprochilodus insignis*.**

| Localities (Rivers) | Water type | Control Region | | | | Microsatellites | | | | | | |
|---|---|---|---|---|---|---|---|---|---|---|---|---|
| | | N | NH | Gene Diversity (± S.D.) | Haplotype Diversity | N | NA | Average gene diversity over loci | $F_{IS}$ ($p$) | $A_{RA}$ | $P_{AR}$ | Average $H_O$–$H_E$ |
| 1. Tabatinga (mainsteam) | white-water | 18 | 14 | 1 ± 0.02 | 0.009 | 17 | 12.4 | 0.88 ± 0.47 | 0.049 (0.0742) | 5.67 | 0.51 | 0.85–0.88 |
| 2. Eirunepé (Jurua River) | white-water | 32 | 25 | 1 ± 0.01 | 0.011 | 20 | 14.4 | 0.91 ± 0.48 | **0.150 (0.0001)** | 5.94 | 0.65 | 0.85–0.90 |
| 3. Tefé (mainsteam) | white-water | 12 | 10 | 1 ± 0.03 | 0.011 | 12 | 9.8 | 0.87 ± 0.47 | 0.021 (0.3395) | 5.42 | 0.40 | 0.85–0.87 |
| 4. Coari (mainsteam) | white-water | 10 | 9 | 1 ± 0.05 | 0.012 | 7 | 7.4 | 0.85 ± 0.48 | **0.165 (0.0004)** | 5.44 | 0.40 | 0.79–0.87 |
| 5. Piagaçu-Purus (Purus River) | white-water | 36 | 27 | 1 ± 0.01 | 0.012 | 19 | 12.6 | 0.87 ± 0.47 | 0.080 (0.0061) | 5.52 | 0.46 | 0.82–0.87 |
| 6. Manaus (mainsteam) | black-water | 21 | 16 | 1 ± 0.01 | 0.012 | 20 | 12.3 | 0.89 ± 0.47 | **0.234 (0.0001)** | 5.73 | 0.46 | 0.76–0.89 |
| 7. Novo Airão (Negro River) | black-water | 23 | 16 | 1 ± 0.01 | 0.009 | 21 | 12.9 | 0.90 ± 0.48 | **0.128 (0.0001)** | 5.75 | 0.43 | 0.82–0.90 |
| 8. Careiro (mainsteam) | white-water | 12 | 7 | 1 ± 0.03 | 0.012 | 9 | 9.0 | 0.90 ± 0.50 | **0.145 (0.0006)** | 5.83 | 0.49 | 0.81–0.90 |
| 9. Santarém (Tapajos River) | clear-water | 32 | 27 | 1 ± 0.01 | 0.010 | 20 | 13.4 | 0.87 ± 0.46 | **0.171 (0.0001)** | 5.68 | 0.42 | 0.76–0.87 |
| 10. Itaituba (Tapajos River) | clear-water | 28 | 20 | 1 ± 0.01 | 0.011 | 18 | 12.0 | 0.88 ± 0.47 | **0.205 (0.0001)** | 5.48 | 0.50 | 0.74–0.87 |
| 11. Jacareacanga (Tapajos River) | clear-water | 17 | 14 | 1 ± 0.02 | 0.010 | 17 | 15.0 | 0.91 ± 0.49 | **0.138 (0.0001)** | 6.09 | 0.85 | 0.82–0.91 |
| All | | 241 | 116 | 0.96 ± 0.01 | 0.011 | 180 | 23.9 | 0.90 ± 0.47 | **0.145 (0.0001)** | 5.81 | 5.81 | 0.81–0.89 |

**Notes.**

N, sample size; NH, Average Number of Haplotypes; NA, Average Number of Alleles (mean calculated by considering the eight microsatellite loci); $F_{IS}$, inbreeding coefficient; $A_{RA}$, Alleles Richness Across loci; $P_{AR}$, Private Allelic richness across loci; $H_O$, observed heterozygosity; $H_E$, expected heterozygosity; S.D., Standard deviation.

Values in bold correspond to significant $P$ values for $F_{IS}$ ($p < 0.00179$).

**Table 3** Analysis of temporal molecular variance (AMOVA) and the $F_{ST}/\Phi_{ST}$ Nm values for comparison of four *S. insignis* localities sampled in different years, obtained through control region and eight microsatellite *loci* in *Semaprochilodus insignis*.

| Markers | Population | $\Phi_{ST}/F_{ST}$ | p values | Nm | Pairwise $\Phi_{ST}/F_{ST}$ | p values |
|---|---|---|---|---|---|---|
| | Tabatinga (2007 *vs* 2008) | 0.22195 | **0.010** | 1752.81 | 0.22195 | **0.013** |
| Control Region | Piagaçu-Purus (2006 *vs* 2007) | −0.01246 | 0.446 | inf | −0.01246 | 0.441 |
| | Santarém (2006 *vs* 2008) | 0.04883 | 0.131 | 9739.71 | 0.04883 | 0.122 |
| | Itaituba (2006 *vs* 2008) | 0.04539 | 0.155 | 10516.77 | 0.04539 | 0.155 |
| | Tabatinga (2007 *vs* 2008) | 0.00267 | 1 | 83898.16 | 0.00592 | 0.330 |
| Microsatellite | Piagaçu-Purus (2006 *vs* 2007) | −0.01260 | 1 | inf | −0.00775 | 0.900 |
| | Santarém (2006 *vs* 2008) | −0.00183 | 1 | 65110.54 | 0.00762 | 0.524 |
| | Itaituba (2006 *vs* 2008) | 0.01429 | 0.937 | 18777.75 | 0.02594 | 0.193 |

**Notes.**
$\Phi_{ST}/F_{ST}$, fixation index.
Values in bold correspond to significant *p* values ($p < 0.05$).

above 0.5 were observed in the locations with the greatest geographical distance (Tabatinga, Eirunepé, Itaituba, and Jacareacanga) (Table 2).

## Temporal and spatial population structure

For temporal analysis based on control region data, the $\Phi_{ST}$ values of AMOVA results (Table 3) ranged from −0.01246 for comparisons between different years in Piagaçu-Purus (2006 × 2007) to 0.22195 for comparisons between different years in Tabatinga (2006 × 2008) with statistical significance for Tabatinga ($p < 0.010$). For this locality 22.19% of variance is found between samples from different years, indicatating temporal structure. The values of $F_{ST}$ of AMOVA for the temporal analysis based in microsatellite data ranged from −0.00183 for Santarém (2006 × 2008) to 0.01429 for Itaituba (2006 × 2008), however, none of the values were significant (Table 3). Likewise, pairwise comparisons for both markers showed low and non significant $F_{ST}$ values with the aforementioned mtDNA Tabatinga (2006 × 2008) comparison (Table 3).

The AMOVA based on the spatial analysis revealed that the higher genetic variation was observed within sampled populations (99.65% for control region and 99.19% for microsatellites), and no genetic differentiation was observed among populations ($\Phi_{ST} = 0.00352$, $p = 0.335$, and $F_{ST} = 0.00813$, $p = 0.937$, for control region and microsatellites, respectively). For the control region the values of pairwise comparisons of $\Phi_{ST}$ for spatial analysis ranged from −0.008 for Novo Airão and Manaus to 0.148 for Tabatinga and Careiro, however, no value was statistically significant after Bonferroni correction ($p < 0.0009$) (Table 4). For the microsatellite data the values of pairwise comparisons of $F_{ST}$ ranged from −0.0003 for Eirunepé and Jacareacanga to 0.037 for Coari and Novo Airão. After Bonferroni correction only the Tabatinga and Careiro ($F_{ST} = 0.032$, $p < 0.0001$) and Tefé and Jacareacanga ($F_{ST} = 0.028$, $p < 0.0001$) comparisons were significant (Table 5). All Nm values were above thousands of effective migrants per generation. When testing population groups as function of different types of water, AMOVA did not show any genetic differentiation ($\Phi_{ST} = 0.00513$, $p = 0.79$, and $F_{ST} = 0.00039$, $p = 1.00$, for control region and microsatellite, respectively).

**Table 4 Pairwise $\Phi_{ST}$ (below the diagonal) and *Nm* values (above the diagonal) among the 11 locations obtained with control region.**

|      | Tab    | Eir      | Tef      | Coa    | PPu      | Mao       | NAi      | Car       | San      | Ita      | Jac      |
|------|--------|----------|----------|--------|----------|-----------|----------|-----------|----------|----------|----------|
| Tab  | –      | 18497.96 | 33241.94 | inf    | 15953.04 | 25360.72  | inf      | 2884.20   | 6646.56  | 6752.21  | inf      |
| Eir  | 0.026  | –        | inf      | inf    | inf      | inf       | inf      | 73848.14  | inf      | inf      | 14936.83 |
| Tef  | 0.015  | −0.042   | –        | inf    | inf      | inf       | inf      | inf       | inf      | inf      | inf      |
| Coa  | −0.005 | −0.045   | −0.061   | –      | inf      | inf       | inf      | inf       | inf      | inf      | 47626.23 |
| PPu  | 0.030  | −0.022   | −0.036   | −0.043 | –        | inf       | inf      | 88813.19  | iinf     | inf      | 17770.61 |
| Mao  | 0.019  | −0.018   | −0.044   | −0.038 | −0.026   | –         | inf      | 142666.67 | 27797.44 | inf      | inf      |
| NAi  | −0.017 | −0.010   | −0.021   | −0.030 | −0.007   | −0.008    | –        | 5333.71   | 30988.01 | 29139.79 | 50577.22 |
| Car  | **0.148** | 0.007 | −0.012   | −0.015 | 0.006    | 0.003     | 0.086    | –         | 62793.08 | inf      | 3593.78  |
| San  | **0.070** | −0.017 | −0.013  | −0.031 | −0.004   | 0.018     | 0.016    | 0.008     | –        | inf      | 4903.86  |
| Itai | **0.069** | −0.016 | −0.023  | −0.032 | −0.011   | −0.004    | 0.017    | −0.019    | −0.014   | –        | 6399.08  |
| Jac  | −0.007 | 0.032    | −0.002   | 0.010  | 0.027    | −0.002    | 0.010    | **0.122** | **0.093** | **0.072** | –        |

Notes.

Tab, Tabatinga; Eir, Eirunepé; Tef, Tefé; Coa, Coari; Ppu, Piagaçu-Purus; Mao, Manaus; NAi, Novo Airão; Car, Careiro; San, Santarém; Ita, Itaituba; Jac, Jacarea-canga.

Bold indicates the significance at the level of $p < 0.05$.

* indicates significance after Bonferroni correction ($p < 0.0009$).

**Table 5 Pairwise $F_{ST}$ (below the diagonal) and *Nm* values (above the diagonal) among the 11 locations obtained with microsatellite markers.**

|      | Tab      | Eir       | Tef       | Coa       | PPu       | Mao       | NAi       | Car       | San       | Ita        | Jac      |
|------|----------|-----------|-----------|-----------|-----------|-----------|-----------|-----------|-----------|------------|----------|
| Tab  | –        | 40297.40  | 43834.00  | 18227.59  | 289088.92 | 48341.49  | 41364.20  | 15313.79  | 77721.90  | 59586.98   | 37015.46 |
| Eir  | **0.012** | –        | 25249.43  | 24066.91  | 30912.42  | 103026.13 | 49264.57  | 28763.06  | 38642.53  | 72261.18   | inf      |
| Tef  | 0.011    | **0.019** | –         | 14709.87  | 139037.53 | 48708.84  | 41010.27  | 16153.75  | 41317.61  | 22519.36   | 17152.57 |
| Coa  | **0.027** | 0.020    | **0.033** | –         | 18338.18  | 51219.52  | 12997.61  | 14183.58  | inf       | 2021197.86 | 16264.48 |
| PPu  | 0.002    | **0.016** | 0.004     | **0.027** | –         | 71825.22  | 78663.51  | 21325.82  | 83505.92  | 48928.68   | 39343.13 |
| Mao  | 0.010    | 0.005     | 0.010     | 0.010     | 0.007     | –         | 40793.63  | 26074.05  | 65002.95  | 48553.49   | 59454.67 |
| NAi  | **0.012** | 0.010    | 0.012     | **0.037** | 0.006     | 0.012     | –         | 40464.39  | 35076.28  | 31107.99   | 61432.94 |
| Car  | **0.032*** | 0.017   | **0.030** | **0.034** | **0.023** | 0.019     | 0.012     | –         | 17834.38  | 20321.69   | 46033.20 |
| San  | 0.006    | **0.013** | 0.012     | −0.001    | 0.006     | 0.008     | **0.014** | **0.028** | –         | 67917.81   | 22602.62 |
| Itai | 0.008    | 0.007     | **0.022** | 0.000     | 0.010     | 0.010     | **0.016** | 0.024     | 0.007     | –          | 35580.70 |
| Jac  | **0.013** | −0.0003  | **0.028*** | **0.030** | 0.013    | 0.008     | 0.008     | 0.011     | **0.022** | 0.014      | –        |

Notes.

Tab, Tabatinga; Eir, Eirunepé; Tef, Tefé; Coa, Coari; Ppu, Piagaçu-Purus; Mao, Manaus; NAi, Novo Airão; Car, Careiro; San, Santarém; Ita, Itaituba; Jac, Jacarea-canga.

Bold indicates the significance at the leval of $p < 0.05$.

* indicates significance after Bonferroni correction ($p < 0.0009$).

The STRUCTURE v.2.3.4 delimited only one biological population among all individuals sampled from the 11 localities, with the highest posterior probability of −7259.2 (Fig. S1, Table S4). These results were also in agreement with the DAPC analysis which also identified only one group present within *S. insignis* (Fig. 3).

The Mantel test results suggest no significant correlation between genetic divergence and geographical distance of sampled localities neither for control region ($r = −0.014751$, $p = 0.505$) nor for microsatellites ($r = −0.267629$, $p = 0.904$).
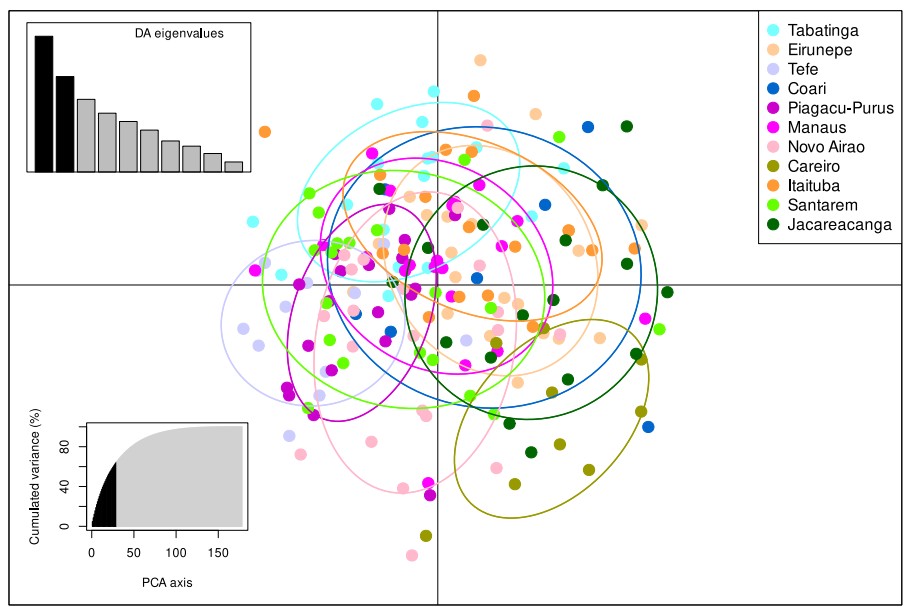

**Figure 3** **DAPC analysis on eight microsatellite loci of 180 individuals of *Semaprochilodus insignis*.** Scatterplots and membership probabilities of the first two principal components of the DAPC analysis on eight microsatellite loci of 180 individuals of *Semaprochilodus insignis*.

### Population sizes and bottleneck tests

The contemporary effective population size ($N_e$) ranged from 61.1 in Itaituba to 491.2 in Novo Airão, with nine locations tending to infinity (Table 6). Bottleneck results based on the standardized differences tests detected significant deviations in observed heterozygosity, suggesting bottleneck signal for *S. insignis* for all localities, with eight localities under the SMM model, and three localities under IAM model. Similarly, the Garza-Williamson index ranged from (0.252–0.420) across populations; all values were lower than the critical value of 0.68 proposed by Garza and Williamson (Table 6) indicating that the populations have experienced a reduction in population size (a bottleneck). Furthermore, bottleneck was also observed when the analysis was conducted considering a single and large population.

## DISCUSSION

Based on the present study, *Semaprochilodus insignis* from the Amazon basin is a large genetically diverse and panmictic population (Table 2). The sampled localities have very similar levels of genetic diversity, and are connected by intense geneflow driven by seasonal migration for feeding and reproduction (Tables 3, 4 and 5). The inbreeding coefficient ($F_{IS}$) was significant for all localities, a characteristic that has been observed in studies carried out with populations of other Amazonian migratory fish (*Oliveira RC et al., 2017*; *Santos, Hrbek & Farias, 2018*; *Escobar, IP & Hrbek, 2022*), which form large shoals, probably composed of small family groups. The migrations of the schools of *S. insignis* are probably not as long as observed for the large migratory catfishes in the Amazon,

**Table 6  Population sizes and bottleneck tests for *Semaprochilodus insignis*.**

| Population | p _stdv_IAM | p _stdv_SMM | Mean Garza-Williamson index | Ne (95% CIs) |
|---|---|---|---|---|
| Tabatinga | 0.177940 | **0.003055** | 0.3189 ± 0.1034 | ∞ (169.7 − ∞) |
| Eirunepé | 0.155051 | **0.011132** | 0.3416 ± 0.0889 | ∞ (54.4 − ∞) |
| Tefé | 0.207798 | **0.042432** | **0.2753 ± 0.0933** | ∞ (27.9 − ∞) |
| Coari | 0.262372 | **0.016668** | **0.2522 ± 0.0883** | ∞ (14.9 − ∞) |
| Piagaçu-Purus | 0.222934 | **0.000216** | **0.2951 ± 0.0699** | ∞ (136.2 − ∞) |
| Manaus | **0.019463** | 0.290048 | **0.2807 ± 0.0567** | ∞ (35.6 − ∞) |
| Novo Airão | **0.009818** | 0.357092 | **0.2826 ± 0.0917** | 491.2 (38.9 − ∞) |
| Careiro | **0.032830** | 0.237083 | **0.2529 ± 0.0667** | ∞ (21.6 − ∞) |
| Santarém | 0.214568 | **0.000033** | 0.3447 ± 0.0916 | ∞ (214.0 − ∞) |
| Itaituba | 0.211736 | **0.000840** | **0.2921 ± 0.0541** | 61.1 (14.0 − ∞) |
| Jacareacanga | 0.235333 | **0.003327** | 0.3383 ± 0.0803 | ∞ (3874.1 − ∞) |
| All | **0.005768** | **0.000000** | **0.4203 ± 0.0632** | 1183.7 (441.8 − ∞) |

**Notes.**

Values in bold correspond to significant values $p < 0.05$

p_stdv, $p$ standard deviation; IAM, Infinite Alleles Model; SMM, Stepwise Mutation Model; CIs, Confidence Intervals.

such as the dourada (*Brachyplatystoma rousseauxii* (Castelnau, 1855)) and the piramutaba (*Brachyplatystoma vaillantii* (Valenciennes, 1840)), but the schools of jaraquis are seen migrating over hundreds of kilometers. Although there has been no previous genetic study to confirm the hypothesis of *S. insignis* forming a single panmictic population in the Amazon basin, this characteristic was expected due to the life style of the species, which migrates extensively through a wide-ranging habitat during its life cycle. *Semaprochilodus insignis* is an r-strategist species which has high reproductive rates, complex schooling behavior, and a migratory and reproductive pattern associated with flooding pulses, typical of the Amazon basin (*Junk, 1997*). The results also corroborate the hypothesis of *Hrbek et al. (2005)* that the seasonality of the floodplains in the Amazon basin and migrations for feeding and reproduction favor the panmictic pattern found in many Amazon fish species along the Amazon River channel, now also including *S. insignis* and many others Amazon fishes such as *Colossoma macropomum* (*Santos, Ruffino & Farias, 2007*; *Farias et al., 2010*; *Santos, Hrbek & Farias, 2018*), *Prochilodus nigricans* (*Machado et al., 2017*), *Brycon amazonicus* (*Oliveira RC et al., 2017*), and *Brachyplatystoma rousseauxii* (*Batista, 2010*). The high degree of genetic diversity appears to be common in migratory and semi-migratory fish that form large panmictic populations. The genetic diversity indexes of *Semaprochilodus insignis* are on part with other Amazonian fishes cited above, and with marine fishes (*DeWoody & Avise, 2000*), both of which have large effective population sizes. This is in contrast to the majority of other freshwater fishes (*Santos, Hrbek & Farias, 2018*; *DeWoody & Avise, 2000*).

Temporal analysis of the population structure of the *S. insignis* in the Amazon basin was carried out in the four cohort pairs in order to determine the possibility of genetically differentiated stocks migrating in different years. The cohorts sampled across different years exhibited no significant differences, which was confirmed by high *Nm*, low $\Phi_{ST}$ and

$F_{ST}$ values based on microsatellites data. With the exception of the Tabatinga temporal samples, mtDNA data also showed no differentiation between the temporal cohorts (Table 3). This suggests that *S. insignis* in the Amazon basin compose a single, large, geographically and temporally panmictic population.

While the temporal samples from the Tabatinga location were significantly differentiated based on the analysis of the mtDNA, we refrain from drawing conclusions based on this result since our sample size for this location was relatively small, and this was the only statistically significant comparison. The results of our temporal analyses are in marked contrast to those observed in the other major South American river drainages, the Parana-Paraguai and Sao Francisco rivers. A number of population genetic studies carried out on taxonomically diverse group of characid fishes suggested the possibility of co-occurrence of genetically differentiated populations in the same drainage system. These species included *Brycon lundii* (*Wasko & Galetti Jr, 2002*), *Prochilodus argenteus* (*Hatanaka, Henrique-Silva & Galetti, 2006*), *Prochilodus costatus* (*Braga-Silva & Galetti, 2016*), and *Salminus brasiliensis* (*Ribolli et al., 2017*). However, until now, no species of fish from the Amazon basin has shown such a pattern.

## Fishing effect and implications for conservation and management

In the central Amazon basin region, the low cost of *Semaprochilodus insignis* is offset by fishery volume (*Faria Jr & Batista, 2019*), making commercial fishery of the *S. insignis* lucrative. The contemporary populations of *S. insignis* show evidence of bottleneck in all analyzed metrics (Table 6), which means that *S. insignis* are indeed suffering from overexploitation. Considering that the alleles are lost more quickly than the heterozygosity, the results based on the analyzes of the Garza-Williamson index (allele loss) and the Bottleneck software (heterozygosity) showed that the reduction of the effective population size has been ongoing for some time (Table 6). The same pattern of the effects of intense fishing was observed in the pirarucu (*Arapaima gigas*), a sedentary species and of great economic value for the Amazon region, and also considered overfished in the rivers of the Amazon basin (*Farias et al., 2019*). The genetic diversity of *S. insignis* is large (Table 2), and thus may lead some to interpret this as lack of effect of overfishing, or that "genetic health" of *S. insignis* has not been affected by overfishing. However, this interpretation is not correct, since genetic diversity per se is not relevant, but rather temporal trends in changes in generic diversity are. While the genetic diversity of *S. insignis* is large, as in other migratory Amazonian fishes with large geographic distributions, we have no baseline, pre-overfishing estimate of genetic diversity, and thus no information on temporal changes in genetic diversity. Since the Garza-Williamson index is based on allele loss and it is significant, *S. insignis* has suffered from loss of diversity despite still having relatively high levels of diversity.

Based on the analyses of 1996 to 2000 fisheries data from the central Amazon, *Vieira (2003)* concluded that all size classes were exploited by the commercial fishery. However, the majority of the catch was concentrated on size classes smaller than 24 cm, a size at which 50% of the individuals reached reproductive maturity. Overfishing and fishing of sub-reproductive individuals consequently has led to selection for reproductive maturity at

smaller size. These observations are consistent with studies of *Batista (1999)* who analyzed fish landing data from 1994 to 1996, and also with *Ribeiro (1983)* who analyzed fisheries data from the late 1970s. All three studies clearly demonstrate that *S. insignis* is overfished.

Given that *S. insignis* is suffering from overexploitation evidenced by (1) decrease of census population size indicated by lower fishing efficiency, (2) its adaptive response of reduction in size at first reproduction (*Batista, 1998*; *Vieira, 2003*; *Batista et al., 2012*), and (3) that it is experiencing a genetic bottleneck, the preservation of the genetic diversity of *S. insignis* must be a high conservation priority. Loss of genetic diversity (effective population size) lags behind the loss of census size, and thus provides a window of opportunity for implementing conservation and management strategies that are able capitalize on the still available genetic diversity.

## Implications for conservation and monitoring

In 2007, IBAMA established a closed fishery period to protect selected overfished species (Ordinance No. 48 of November 5, 2007). The closed period aims to close fishing during the reproductive season, in order to guarantee the reproduction of the selected species and help in the maintenance of fish stocks. However, despite clear evidence of overexplitation detected as early as 40 years ago (*Ribeiro, 1983*; *Batista, 1999*; *Vieira, 2003*), *S. insignis* fishing regulations have been implemented in a very irregular way, if at all, and independently by each region of the state of Amazonas (http://www.ipaam.am.gov.br/defeso/, accessed on January 3, 2023). Rather, the entire fishery focuses on harvesting reproductive migrations. Considering the results presented here, *S. insignis* is clearly showing signs of rapid demographic population collapse and loss of genetic diversity. Compounded on the overfishing, loss of genetic diversity and lack of fishing regulations, a study by *Ramos (2016)* demonstrated that deforestation in the varzea and igapo forests will negatively affect the reproductive success of *S. insignis*. The varzea and igapo forests are floodplain forests and integral part of the riverine ecosystem, but also are the main area of human settlement and agricultural conversion and unsustainable resource extraction.

The situation of *S. insignis* appears to be analogous to that of the north Atlantic cod (*Gadus morhua*). The north Atlantic cod, a historically abundant r-strategist, has been subject to sustained but sustainable fishery since the Viking period (*Sodeland et al., 2022*). After WWII, as fishing became industrialized, catches peaked despite ever increasing effort to yield ratio, resulting in a partial (60–70%) collapse in the early 1970's. Despite repeated warnings from the scientific community, the fisheries were unsustainably managed, leading to a sudden collapse in 1992, when the north Atlantic cod population dropped to 1% of its historic levels (*Hamilton & Butler, 2001*). Thirty years later, the north Atlantic cod has yet to recover.

The scenario of *S. insignis* appears not to be different. Diverse indicators, whether biological or fisheries, indicate a stressed, overfished and unustainably harvested species on the brink of a demographic collapse. Therefore it is essential that *S. insignis* is placed on the list of species for which fishing is closed during the reproductive season (November 15 to March 31). This will not only decelerate observed negative population trends, but also

provide time during which managers and scientists can devise and implement an effective conservation and management plan.

## CONCLUSIONS

This study is the first to explore the population genetics of *S. insignis*, one of the most important fish species in the Amazon. Here, populations distributed in 11 localities along of Amazon River and their tributaries were analyzed from mitochondrial and microsatellites markers providing information about genetic diversity, temporal and spatial structure and effective population sizes. The results reveal that *S. insignis* forms a single panmitic population, with high levels of genetic variability and gene flow in the Amazon basin. Despite high levels of genetic variability, *S. insignis* shows signs of bottlenecks indicating that the species is suffering the effects of overfishing. Therefore, we hope that the results presented in this study will contribute to the elaboration and implementation of future management plans and fishing regulations. Last but not least, we strongly recommend that the species enters on the list of species protected by fishing closure during its reproductive season.

## ACKNOWLEDGEMENTS

We thank the IPAAM (Instituto de Proteção Ambiental do Amazonas) staff for providing the information for the anual closed fishing data reported here. This study forms a portion of a dissertation of KB at the Biological Diversity program of UFAM, and a portion of a monograph of ISN at the Biological Science undergraduate program at UFAM.

### Funding
This work was supported by grants from CNPq/PPG7 557090/2005-9 and CNPq/CT-Amazônia 575603/2008-9 to I.P.F. The funders had no role in study design, data collection and analysis, decision to publish, or preparation of the manuscript.

### Grant Disclosures
The following grant information was disclosed by the authors:
CNPq/PPG7: 557090/2005-9.
CNPq/CT-Amazônia: 575603/2008-9.

### Competing Interests
The authors declare there are no competing interests.

### Author Contributions
- Ingrid Nunes conceived and designed the experiments, performed the experiments, analyzed the data, prepared figures and/or tables, authored or reviewed drafts of the article, and approved the final draft.

- Kelmer Passos performed the experiments, prepared figures and/or tables, authored or reviewed drafts of the article, and approved the final draft.
- Aline Mourão Ximenes performed the experiments, prepared figures and/or tables, authored or reviewed drafts of the article, and approved the final draft.
- Tomas Hrbek conceived and designed the experiments, analyzed the data, prepared figures and/or tables, authored or reviewed drafts of the article, and approved the final draft.
- Izeni Pires Farias conceived and designed the experiments, analyzed the data, prepared figures and/or tables, authored or reviewed drafts of the article, and approved the final draft.

### Animal Ethics

The following information was supplied relating to ethical approvals (*i.e.*, approving body and any reference numbers):

Instituto Brasileiro do Meio Ambiente e dos Recursos Naturais Renováveis (IBAMA/SISBIO permit no. 11325-1).

### Data Availability

The sequence data are available at GenBank: ON972834–ON973074.

The raw genotype data is available at: Nunes, Ingrid de Souza, Passos, Kelmer Batalha, Ximenes, Aline Mourão, Hrbek, Tomas, & Farias, Izeni Pires. (2023). Spacial and temporal genetic pattern of Semaprochilodus insignis (Prochilodontidae), the most popular fish from the Amazon basin (Version 1) [Data set]. Zenodo. https://doi.org/10.5281/zenodo.7641680

Both sets of data are also available in the Supplemental Files.

### Supplemental Information

Supplemental information for this article can be found online at http://dx.doi.org/10.7717/peerj.15503#supplemental-information.

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
