# Peer review of "Spatial and temporal population genetic analysis of Semaprochilodus insignis (Prochilodontidae), an overexploited fish from the Amazon basin"

_PeerJ, doi:10.7717/peerj.15503_

## Round 0.1 · original submission · Major Revisions

The work needs major changes before it can be considered again. An updated literature review is essential to improve the introduction and discussion with current citations. In addition, some of the results are based on a low number of samples, which raises doubts about the results obtained. In case of not being able to have data from more specimens, it is recommended to state this fact in the discussion itself.

Reviewer 1 ·

Basic reporting

The manuscript assesses population genetics of an Amazonian fish species by sequencing the control region and genotyping eight microsatellite loci from individuals collected in 11 sampling sites in the Amazon River basin. The results revealed that the species constitutes a single panmitic population, with high levels of genetic variability and gene flow in the Amazon basin. Bottleneck was detected in some local populations and the authors suggest that this could be a genetic signature of overfishing. They suggest their results will contribute to management and conservation of this fish.
A major point is of note; if the authors found no spatial population structuring why they considered the local populations independently for variation genetic calculation; it would be more understandable if all variation genetic analyses were conduced as a single and large population. In this way, they would avoid to using very small sample as representing a local population (as Coari, Tefé, Careiro, for instance). I would like to see the genetic analyses for a single population as they found; what about bottleneck, it would be seen in a single population? If so, it would reinforce their conclusion on linking bottleneck sign with overfishing.
I felt weak how the authors explained the results of population expansion “This result provides evidence of population expansion, meaning that these populations are either experiencing or have recently experienced population growth, which is very promising.” It looks contradictory to overfishing they argue above in the text. Moreover, mtDNA signatures is probably more ancient than recent one.
“As the analyses of the present work were carried out in populations from almost two decades ago, it would be interesting to carry out further comparative analyzes to verify how populations reacted to these regulatory measures.” It is unlikely to see any mtDNA signatures in two decades.
A careful revision across the references is needed. Several cited references were not listed or were listed but not cited (see below). Several references are thesis or documents not indexed, should be avoided.
References cited in the main text but not included in reference list:
Neto & Dias, 2015; Neto Dias (2015); Hrbek & Farias (2008); Van Oosterhout, Hutchinson, Wills & Shipley, 2004; Do et al. 2013; Waples & Do 2008; Piry et al. 1999; Garza and Williamson 2001; Junk, 1997; Farias et al., 2010; Santos et al., 2018; Machado et al., 2016; Oliveira et al., 2017; Riboli et al., 2016; Batista, 2000; Farias et al., 2019; Batista (1999); Ribeiro (1983); Wright, 1951
References of listed but not cited in the main text:
Batista, J. S., & Alves-Gomes, J. A. (2006). Phylogeography of Brachyplatystoma rousseauxii (Siluriformes - Pimelodidae) in the Amazon Basin offers preliminary evidence for the first case of <homing= for an Amazonian migratory catfish. Genetics and Molecular Research, 5(4), 7233740.
Farias, I. P., & Hrbek, T. (2008). Patterns of diversification in the discus fishes (Symphysodon spp. Cichlidae) of the Amazon basin. Molecular Phylogenetics and Evolution, 49(1), 32343. http://doi.org/10.1016/j.ympev.2008.05.033
Freitas, C. E. de C., & Rivas, A. A. F. (2006). A pesca e os recursos pesqueiros na Amazônia Ocidental. Ciência E Cultura, 58(3), 30332. http://doi.org/ISSN 2317-6660
Galletti, E. S. (2009). Distribuição da variabilidade genética da pescada, Plagioscion squamosissimus (Heckel,1840) na calha do rio Amazonas. Dissertação (Mestrado em Genética, Conservação e Biologia Evolutiva). Instituto Nacional de Pesquisas da Amazônia. Manaus, 67 pp.
Hrbek, T., Crossa, M., & Farias, I. P. (2007). Conservation strategies for Arapaima gigas (Schinz, 1822) and the Amazonian várzea ecosystem. Brazilian Journal of Biology = Revista Brasleira de Biologia, 67(4 Suppl), 9093917. http://doi.org/10.1590/S1519-69842007000500015
Passos, K. B. (2009). Genética populacional do jaraqui de escama grossa (Semaprochilodus insignis - Prochilodontidae, Characiformes). Dissertação (Mestrado em Diversidade Biológica) Universidade Federal do Amazonas, Manaus.
Rodrigues, F. C. (2009). Estimativa da variabilidade genética da piramutaba (Brachyplatystoma vaillantii) por meio de marcadores moleculares microssatélites e D-loop de quatro localidades da Amazônia: diferenças entre calha e tributários. Dissertação (Mestrado em Genética Conservação e Biologia Evolutiva). Instituto Nacional de Pesquisas da Amazônia. Manaus, 96 pp.
Ruffino, M. L. (2004). A pesca e os recursos pesqueiros na Amazônia brasileira. IBAMA/ProVárzea.
Other minor points:
1) Full scientific name (Semaprochilodus insignis) must appear once in the text (as in line 46, Introduction section); after that it should be S. insignis.
2) Line 110 – delete “Add your introduction here.”
3) Line 122 – ….able to collect…..
4) “software” instead of “program” where it occurs in the text; for instance, the BioEdit software
5) Line 160. Re-draft “SinD03The PCR reaction of …….” And delete an “of”
6) Line 184. Better FIS than FIS
7) Line 362. C. macropomum

Experimental design

See above

Validity of the findings

See above

Additional comments

See above

Reviewer 2 ·

Basic reporting

Materials and Methods: English language should be criteriously revised and improved.
Current PCR reactions descriptions are difficult to understand. I suggest use only the final volume and final concentration for each reagent this will improve the clarity.

Figures
Figure 1 - I suggest indicating which sampling locations have black, white, and clear waters.
Figure 2 - Sample Locations legends are difficult to read. I suggest using larger font.

Tables
Table 3 - The information yn the last two columns are redundant with information about Fst and 0st.
Table 4 - Legends of table are incorret (Inf and % descriptions are not part of this table). I couldn´t understand the diference between table 3 and 4. Why the values of 0st (control region) are the same of table 3 and all other values are different?
Once the divergences are explained, I still suggest joining the contents of the two tables.

Table 5 - Unnecessary table, the content has already been metioned in the text

Minor revisions
Line 84: a dot out of place
Line 110: the phrase - Add your introduction here – should be deleted.
Line 260 - GenBank access numbers were not described.
References - some references has DOI, others do not. Please standardize

Experimental design

Methods - Should be improved, as mentioned before, to achieve more clarity.

I have some concerns about some locations sample size. As told by the author, the specie seems to be a panmitic specie with a large population. In this way, I questioned if the sample size smaller then 20 individuals are enough to make any assumptions or inferences.
In the same way, the temporal structured analysis seems to be inadequate since the collection sites Tabatinga and Piagaçu-Purus, for example, have only 6 individuals in the year 2008/2007 respectively. It is possible that the differences observed between the 2007 and 2008 Tabatinga sampling are due to the bias of the small sample number.

Still on the temporal structured analysis , the sampling years compared are not the same (Tambatinga 2007 x 2008; Piagaçu 2006 x 2007, Santarém 2006 x 2008; and Itaituba 2006 x 2008). If the authors wants to interrogate the possibility of genetically differentiated stocks migrating in different years, the sampling year shoud be padronized.

It is unclear whether the samples used for the microsatellite analyzes are the same (subsamplings, since "n" is smaller) as those used for the control region. If there are not the same, it is not possible to say that the results of one corroborate with the other.

Validity of the findings

Line 394-395 and 417 - 418 - it is not clear why the authors claim that the species undergoes a bottleneck - this was not explained.
Additionally, there is no evidence of a sharply reduction of population size (botleneck premise) to make this statement (conclusions

Additional comments

The subject is interesting and has much to contribute to the basic studies of the S. insignis. I suggest that the temporal analyzes be removed from the article, without prejudice to the robustness of the manuscript.
Further clarifications should be given concerning the sample size, especially for analyzes with MS.

Reviewer 3 ·

Basic reporting

The article is written in clear, unambiguous and technically correct English text. The article complies with professional standards of courtesy and expression.

The article's introduction should present a paragraph with the main genetic findings obtained for the species or genus in the region. I also think that the discussion should be based on more up-to-date references and research on population genetics of commercial species in more recent studies.

The article structure conforms to an acceptable standard format but the Legends of figures and tables may be more explanatory. In some, the information on which species the data refers to is lacking. Or lack information about which locality or region.

All appropriate raw data has been made available in accordance with the Data Sharing policy.

In the present study, the authors consider the historical situation of overfishing, however, considering the period of overexploitation from 1995 to 2010, with samples obtained between 2006 to 2008, it is not a historical period between the large volume of landings and the data analyzed. If we were considering current data with overexploitation events it would be correct to consider the 'historical situation'. Therefore, I think the data are very important and relevant, but I understand that this is not a historical situation from that period.
Unless there were current data from the last few years, comparing historical data from 2006 to 2008.
Overexploitation events from 1995 to 2010 may not have been detected in the population from 2006 to 2008 due to the proximity of time.
But evidently the data are important as a basis for evaluating the dynamics of population genetic variation over time.

Experimental design

This is an original primary Research within the Objectives and Scope of the journal.

The submission clearly defines the research question, which is relevant and significant. There really is a knowledge gap that is being investigated and the study presents necessary approaches to contribute to filling this gap.

The investigation was conducted rigorously and with a high technical standard. The research was conducted in accordance with the ethical standards in force in the area.

The methods were described with enough information to be reproduced by another investigator.

Validity of the findings

The study is relevant to scientific literature and contributes to the knowledge and action plans of the ichthyofauna species in the Amazon region, especially for species of commercial value with interest and concern about overfishing in the Amazon basin.

The data on which the conclusions are based were provided or made available in a specific repository in the area of ​​genetic studies (GenBank). The data are robust, statistically sound and controlled.

The conclusions were properly formulated, being related to the original question investigated and limited to those supported by the results.

Additional comments

I think it is very important that recommendations are made so that managers and researchers can direct better plans for fisheries control and protection of biological, ecological and fisheries resources in the region.

Annotated reviews are not available for download in order to protect the identity of reviewers who chose to remain anonymous.

---

## Round 0.2 · Minor Revisions

Many thanks for the work developed in order to improve the manuscript. There are still some modifications needed, as noted by Reviewer 2:

Table 6 - Legends are incomplete. The acronyms are not explained

Line 180: to correct FISFis

Lies 423 to 442 – the end of the discussion is about salmon without mentioning the S. insignis. This part should be removed once there is no correlation to the subject.
Thanks you very much in advance.

Reviewer 2 ·

Basic reporting

Almost all the recommendations were observed.
Some remaining minor revisions:

Table 6 - Legends are incomplete. The acronyms are not explained
Line 180: to correct FISFis
Lies 423 to 442 – the end of the discussion is about salmon without mentioning the S. insignis. This part should be removed once there is no correlation to the subject.

Experimental design

Ok

Validity of the findings

ok

Additional comments

ok

Reviewer 3 ·

Basic reporting

No comment

Experimental design

No comment

Validity of the findings

No comment

Additional comments

No comment

---

## Round 0.3 · accepted · Accept

Thank you for submitting your work to this journal.